# Constrained Planar Array Thinning Based on Discrete Particle Swarm Optimization with Hybrid Search Strategies

**DOI:** 10.3390/s22197656

**Published:** 2022-10-09

**Authors:** Wanhan Cai, Lixia Ji, Chenglin Guo, Ke Mei, Hao Zeng

**Affiliations:** School of Microelectronics and Communication Engineering, Chongqing University, Chongqing 400044, China

**Keywords:** array thinning, search strategy, peak side-lobe level, particle swarm optimization

## Abstract

This article presents a novel optimization algorithm for large array thinning. The algorithm is based on Discrete Particle Swarm Optimization (DPSO) integrated with some different search strategies. It utilizes a global learning strategy to improve the diversity of populations at the early stage of optimization. A dispersive solution set and the gravitational search algorithm are used during particle velocity updating. Then, a local search strategy is enabled in the later stage of optimization. The particle position is adaptively adjusted by the mutation probability, and its motion state is monitored by two observation parameters. The peak side-lobe level (PSLL) performance, effectiveness and robustness of the improved PSO algorithm are verified by several representative examples.

## 1. Introduction

Thinned arrays have been the focus of research in recent years for their lower cost, lower energy consumption and lighter weight compared with the conventional uniform arrays. The main purpose of array thinning is to obtain a lower peak side-lobe level (PSLL) on the condition that the antenna array satisfies gain demand. Planar array thinning can be achieved by adjusting the “ON” or “OFF” states of each element in a uniform array.

To suppress the PSLL, several optimization methods have been proposed. As suggested by Liu in [1], the thinning of a reconfigurable linear array can be reduced by matrix beam method with high efficiency. RF switching technology in the T/R module makes the application of adaptive sparse technology possible [2]. Rahardjo et al. [3] developed a new method for designing linear thinned arrays using spacing coefficients based on Taylor line source distribution, which is verified to be able to correctly match different beam pattern configurations without repeated global optimization. Recently, analytical thinning methods by convex optimization [4] and Bayesian compressive sensing [5] have been introduced, but these methods need to set a suitable reference pattern as a precondition. Keizer [6] proposed the iterative Fourier technique (IFT) for array thinning. He calculates the array factor and excitation of the uniform array by making use of the Fourier transforms. Moreover, the IFT method is also used to optimize large planar arrays with higher convergence speed [7].

Benefitting from excellent global search performance, some intelligent optimization algorithms, such as the real genetic algorithm (RGA) [8] and asymmetric mapping method with differential evolution (DE) [9], also play a good role in array thinning. For this problem, analyses are also carried out with the newly introduced Honey Badger Algorithm (HBA) and Chameleon Swarm Algorithm (CSA) algorithms in [10]. Vankayalapati introduced a new algorithm called multi-objective modified binary cat swarm optimization to deal with the optimization of multiple contradicting parameters [11].

First proposed by Eberhart and Kennedy in 1995 [12], particle swarm optimization (PSO) is a new optimization algorithm for solving continuous and discontinuous problems in multiple dimensions. With a variety of analytical and numerical tools available, PSO has been applied to different application domains such as antenna pattern synthesis, array thinning, and sensor networks. The PSO and its improved algorithms are employed frequently for suiting multiple objectives simultaneously. Random drift particle swarm optimization (RDPSO), which is used to solve the electric power economic scheduling problem, has been applied to the optimal design of thinned array in [13], achieved good results. To compare the performance of single-ring planar antenna arrays, Bera et al. [14] proposed a novel wavelet mutation-based novel particle swarm optimization (NPSOWM). By applying the binary particle swarm optimization (BPSO) algorithm to the total focusing method (TFM) [15] for thinned array design, the simulation results indicate that the proposed TFM can greatly increase computational efficiency and provide significantly higher image quality. The PSO study in [16] aims to generate multiple-focus patterns and a large scanning range for random arrays thinning, which is applied to the ultrasound treatment of brain tumors and neuromodulation. A search mode with multi-objective particle swarm optimization was proposed in [17] to solve the problem of optimal array distribution, which met the requirements of reducing the number of antenna elements and maintaining the PSLL simultaneously. In 2021, Guo et al. [18] studied the SLL reduction optimization of linear antenna arrays (LAA) with specific radiation characteristics and circular antenna arrays (CAA) with isotropic radiation characteristics. In the optimization process, an optimization method (GWO-PSO) combining gray wolf optimization and PSO will be used.

In general, in order to avoid the rapid loss of particle distribution in solution space, the existing PSO methods give a variety of evaluation strategies for the diversity of particle population distribution. However, these methods mainly focus on improving the algorithm efficiency and ignore the balance between global search and local search, resulting in the lack of ability to adjust the search focus dynamically in different search stages.

In this paper, a new novel optimization algorithm for large array thinning is proposed. The innovative part of this algorithm is the combined usage of different particle learning strategies with discrete particle swarm optimization, which enhances the ability of global search and effectiveness of the algorithm.

The rest of this paper is organized as follows: The planar array structure and optimization problem model are briefly outlined in Section 2. Section 3 introduces the DPSO algorithm and our improvement to it. Several simulation results and discussions are presented in Section 4. Finally, a brief conclusion is given in Section 5.

## 2. Optimization Model

Assume a large planar array with elements arranged in square grids with a spacing of *d* along *M* columns and *N* rows, as shown in Figure 1.

Matrices ***A*** and ***B*** are set as:(1)$$A=\left(\begin{array}{llll} a_{00} & a_{01} & \cdots & a_{0N-1}\\ a_{10} & a_{11} & \cdots & a_{1N-1}\\ \vdots & \vdots & \ddots & \vdots \\ a_{M-10} & a_{M-11} & \cdots & a_{M-1N-1} \end{array}\right)$$
(2)$$B=\left(\begin{array}{llll} b_{00} & b_{01} & \cdots & b_{0N-1}\\ b_{10} & b_{11} & \cdots & b_{1N-1}\\ \vdots & \vdots & \ddots & \vdots \\ b_{M-10} & b_{M-11} & \cdots & b_{M-1N-1} \end{array}\right)$$
where *a_mn_* and *b_mn_* are the excitation of element (*m*, *n*) and the “ON” or “OFF” state of the element (*m*, *n*), respectively. So, *b_mn_* is 1 or 0. In Figure 1, *θ* and *φ* are the elevation and the azimuth angles in the spherical coordinate, respectively. *u* and *v* are direction cosines defined by *u* = sin *θ* cos *φ* and *v* = sin *θ* sin *φ*. If (*u*_0_, *v*_0_) is the desired main beam direction, the radiation beam pattern *F*(*u*, *v*) can be expressed as:(3)$$F\left(u,v\right)=\sum_{m=0}^{M-1}\sum_{n=0}^{N-1}a_{mn}b_{mn}e^{jk\left[nd\left(u-u_{0}\right)+md\left(v-v_{0}\right)\right]}$$

The fitness function considered in the present study is the PSLL of the radiation beam pattern, which is desired to be as low as possible. The PSLL of the planar array can be computed as:(4)$$\mathrm{PSLL}=\underset{\left(u,v\right)\in S}{\max}\left[20\mathrm{lg}\left|\frac{F\left(u,v;u_{0},v_{0}\right)}{F\left(u_{0},v_{0}\right)}\right|\right]$$
where *S* denotes the angular region excluding the main beam. Considering the constraint of the array filling rate, the sum of all elements in matrix ***B*** should be a definite constant. If the aperture remains the same, the four corner elements of the planar array must be “ON”. The model of optimization can be represented as:(5)$$\left.\begin{array}{c} \underset{B}{\min}\left(\mathrm{PSLL}\right)\\ s.t.0\leq m\leq M-1,0\leq n\leq N-1\\ b_{00}=1,b_{M-10}=1\\ b_{0N-1}=1,b_{M-1N-1}=1\\ \sum_{n=0}^{N-1}\sum_{m=0}^{M-1}b_{mn}=K \end{array}\right\}$$
where *K* denotes the number of elements that turned “ON”.

## 3. Improved Algorithm

This section may be divided by subheadings. It should provide a concise and precise description of the experimental results, their interpretation, as well as the experimental conclusions that can be drawn.

### 3.1. Fundamental PSO Algorithm

The fundamental PSO algorithm assumes a swarm of particles in the solution space, and the positions of these particles indicate possible solutions to the variables defined for a specific optimization problem. The particles move in directions based on update equations impacted by their own local best positions and the global best position of the entire swarm.

Assume a swarm composed of NP particles is uniformly dispersed in a *D*-dimensional solution space, where the position ***x****_i_* and velocity ***v****_i_* of *ith* particle can be expressed as:(6)$$x_{i}=\left(x_{i1},x_{i2},\cdots ,x_{iD}\right)$$
(7)$$v_{i}=\left(v_{i1},v_{i2},\cdots ,v_{iD}\right)$$

The velocity update equation is given below:(8)$$v_{ij}\left(t+1\right)=w\cdot v_{ij}\left(t\right)+c_{1}r_{1}\left(t\right)\left[pBest_{ij}\left(t\right)-x_{ij}\left(t\right)\right]+c_{2}r_{2}\left(t\right)\left[gBest_{j}\left(t\right)-x_{ij}\left(t\right)\right]$$
where *i* = 1, 2, …, *NP*, *j* = 1, 2, …, *D*. Obviously, *NP* represents the number of particles, and *D* represents the number of dimensions. *c*_1_ and *c*_2_ are the acceleration constants, and *r*_1_ and *r*_2_ are two random numbers within the range [0,1]. *v_ij_*(*t*) and *x_ij_*(*t*) are the velocity and position along the *j^th^* dimension for the *i^th^* particle at *t^th^* iteration, respectively. *pBest_ij_*(*t*) is the best position along the *j^th^* dimension for the *i^th^* particle at *t^th^* iteration, also called “personal best’’. Finally, “global best’’ *gBest_i_*(*t*) is the best position found by the swarm along the *j^th^* dimension at *t^th^* iteration.

The value of a position along each dimension for each particle is limited to “0” and “1” in discrete algorithms. The position update equation along the *j^th^* dimension for the *i^th^* particle is given by
(9)$$s\left(v_{ij}\right)=\frac{1}{1+e^{-v_{ij}}}$$
(10)$$x_{ij}\left(t+1\right)=\left\{\begin{array}{l} 1,\begin{array}{cc} & r<s\left(v_{ij}\right) \end{array}\\ 0,\begin{array}{cc} & others \end{array} \end{array}\right.$$
where *s*(*v_ij_*) denotes a function that maps the particle velocity to the probability of the particle position, and *r* represents a random number within the range [0,1].

### 3.2. DPSO with Hybrid Search Strategies

The improved PSO algorithm proposed in this paper consists of three main strategies, which are the global learning strategy based on niche technique, dispersed solution sets, the local mutation search strategy and the motion state monitoring strategy. The execution of the corresponding strategies is adaptively adjusted in different search stages of the optimization.

#### 3.2.1. Global Learning Strategy

It is necessary to maintain as high a swarm diversity as possible in the early stage of optimization to avoid premature convergence. So, two strategies are utilized in the early stage, which are the niche technique and the gravitational search algorithm.

The niche technique proposed in [19] can form multiple stable subpopulations within a population. Each particle interacts only with its immediate neighbors through a ring topology. Obviously, the ring topology structure can promote individuals to search thoroughly in its local neighborhood before global optimization. Therefore, it is acceptable for discovering multiple optima. On the basis of the ring topology proposed by [19], a high-quality solution set P*_G_* is proposed in this paper, as shown in Figure 2. The solution set P*_G_* consist of two kinds. One is the optimal solutions of all particles, the other is the eliminated optimal solutions of some particles with excellent fitness function values. Preserving the possibility of interaction between each particle and its neighbors, particles can also learn from P*_G_* directly. This structure will substitute for the role of the global best position *gBest* in (8).

Because the positions of elements in P*_G_* may be uniformly distributed in the whole solution space, particles under the guidance of elements in PG have a considerable number of potential motion directions, which improves the diversity of the population, and avoids premature convergence.

The gravitational search algorithm (GSA) mentioned in [20] is a novel heuristic search algorithm, where a particle is impacted by the combined gravity of all the other particles in the solution space. We use a GSA-based acceleration to replace the personal best position *pBest_i_* in (8) to enhance the correlation between particles.

At the *t^th^* iteration, the gravitational attraction of particle *q* on particle *i* in the *k^th^* dimension can be defined as:(11)$$F_{iq}^{k}\left(t\right)=G\left(t\right)\frac{M_{i}\left(t\right)\times M_{q}\left(t\right)}{R_{iq}+\varepsilon }[x_{q}^{k}\left(t\right)-x_{i}^{k}\left(t\right)]$$
where *M_q_*(*t*) represents the inertial mass of the particle applying force, *M_i_*(*t*) represents the inertial mass of the particle subjected to the force, *R_iq_* = ||***x****_i_*(*t*) − ***x****_q_*(*t*)||, *R_iq_* denotes the Euclidean distance between particle *i* and particle *q*, and *ε* is a small constant to make sure the denominator is not zero. *G*(*t*) is the gravitational constant whose value changes dynamically, which can be expressed as:(12)$$G\left(t\right)=G_{0}e^{-\alpha \frac{t}{T}}$$
where *α* represents the attenuation rate, *T* is the max iteration times, and *G*_0_ is the initial value.

At the *t^th^* iteration, the resultant force $F_{i}^{k}\left(t\right)$ caused by all the other particles will play a role to *i^th^* particle in the *k^th^* dimension, so:(13)$$F_{i}^{k}\left(t\right)=\sum_{q=1,q\neq i}^{NP}rand_{q}\cdot F_{iq}^{k}\left(t\right)$$
where *rand_q_* is a random number within the range [0,1].

According to Newton’s second law, the acceleration produced by this resultant force can be expressed as:(14)$$a_{i}^{k}\left(t\right)=\frac{F_{i}^{k}\left(t\right)}{M_{i}\left(t\right)}$$

The inertial mass of each particle is calculated from its fitness function value and the updating equation of inertial mass is given below:(15)$$\left\{\begin{array}{l} m_{i}\left(t\right)=\frac{{\mathrm{PSLL}}_{i}\left(t\right)-gWorst\left(t\right)}{gBest\left(t\right)-gWorst\left(t\right)}\\ M_{i}\left(t\right)=\frac{m_{i}\left(t\right)}{\sum_{q=1}^{NP}m_{q}\left(t\right)} \end{array}\right.$$
where PSLL*_i_*(*t*) represents the fitness function value of particle *i* at *t^th^* iteration, and *gWorst*(*t*) is the worst position found by the swarm at *t^th^* iteration.

Therefore, after applying the global learning strategy, the updating equation of particle velocity can be rewritten as:(16)$$v_{id}\left(t+1\right)=\left\{\begin{array}{l} w\cdot v_{id}\left(t\right)+c_{1}r_{1}\left(t\right)\cdot a_{id}\left(t\right)\begin{array}{cc} , & d\notin C \end{array}\\ w\cdot v_{id}\left(t\right)+c_{1}r_{1}\left(t\right)\left[pGood_{id}\left(t\right)-x_{id}\left(t\right)\right]\begin{array}{cc} , & d\in C \end{array} \end{array}\right.$$
where *d* = 1, 2, …, *D*, *D* represents the number of dimensions, *C* denotes a set of *q* dimensions randomly selected from all *D* dimensions, *pGood_id_*(*t*) is the best position of *l* candidate solution randomly selected from P*_G_* and neighborhood particles along the *d^th^* dimension for the *i^th^* particle at *t^th^* iteration, *a_id_*(*t*) is the acceleration along the *d^th^* dimension for the *i^th^* particle at *t^th^* iteration.

#### 3.2.2. Local Search Strategy

An algorithm should have strong local search ability to improve the convergence efficiency in the later stage of optimization, especially when it is applied to the optimal design of large-scale array. The flowchart of the local search strategy is shown in Figure 3.

It can be considered that a particle *i* can be regarded as moving from a certain position ***x****_i_* to another position ***x^’^****_i_* in the solution space when ***x****_i_* along some dimensions are mutated. Therefore, we propose a local search method that the neighborhood of the optimal position of a particle *i* is searched by only changing partial elements of its position ***x****_i_* under the guidance of a Gaussian random variable. The position moving the equation along the *j^th^* dimension for the *i^th^* particle is given by:(17)$$x_{ij}=\left\{\begin{array}{l} -x_{ij},\;r_{j}<\left|{Χ}_{ij}\left(0,\sigma^{2}\right)\right|\\ x_{ij},\;r_{j}\geq \left|{Χ}_{ij}\left(0,\sigma^{2}\right)\right| \end{array}\right.$$
where *X_ij_* (0, *σ*^2^) represents a Gaussian random variable with a mean value of 0 and standard deviation of *σ*, and *r_i_* is a random number within the range [0,1].

The mutated position 
$x_{i}^{\prime }$
should be used to replace ***x****_i_* if it has a better fitness function value, and the standard deviation *σ* should be amplified to increase the moving distance of particles. Otherwise, keep ***x****_i_* unchanged and reduce *σ*. The updating equation of standard deviation *σ* can be expressed as:(18)$$\sigma \left(k+1\right)=\left\{\begin{array}{l} \sigma \left(k\right)\cdot \rho ,\;\mathrm{PSLL}\left(x_{i}^{\prime }\right)<\mathrm{PSLL}\left(x_{i}\right)\\ \sigma \left(k\right)\cdot \mu ,\;\mathrm{PSLL}\left(x_{i}^{\prime }\right)\geq \mathrm{PSLL}\left(x_{i}\right) \end{array}\right.$$
where *k* represents the number of variations, *ρ* is the expansion parameter and *ρ* > 1, *μ* is the contraction parameter and 0 < *μ* < 1. If the size of the standard deviation *σ* is less than a preset threshold *σ_e_*, that means there is no better solution for particle *i* in the adjacent position. So, the update should be stopped.

#### 3.2.3. Particle Movement Condition Monitoring

In order to avoid some particles falling into local optimum prematurely and to improve the performance of the optimization algorithm, a condition monitoring strategy is proposed to adjust the position of the particle according to the moving state of the particle.

There are two preconditions for a particle *i* to trigger a mutation operation. The first one is that the optimal solution of particle *i* has not been updated for *u* iterations, which can be expressed as:(19)$${\mathrm{Tpb}}_{i}>u$$
where Tpb*_i_* represents the number of iterations of the optimal solution that have not been updated. The second precondition is that the moving distance of the position ***x****_i_* is less than the set value *ε* for successive *h* iterations, which can be expressed as:(20)$${\mathrm{Txb}}_{i}>h$$
where Txb*_i_* represents the number of iterations in which the moving distance of the position ***x****_i_* is less than the set value *ε*. The moving distance is defined as the number of different elements in all dimensions before and after the particle position changes.

It can be assumed that particle *i* has fallen into local optimum when both preconditions are met, and then an individual variation probability *γ_a_* is used to vary all dimensions of position ***x****_i_*. The equation of variation can be represented as:(21)$$x_{id}=\left\{\begin{array}{l} -x_{id},\;r_{d}<\gamma_{a}\\ x_{id},\;r_{d}\geq \gamma_{a} \end{array}\right.$$
where *d* = 1, 2, …, *D*, *D* represents the number of dimensions, and *r_d_* is a random number within the range [0,1].

### 3.3. Steps of Algorithm

The new Algorithm 1 exploits the hybrid search strategies (HSS) described above to improve the performance of a fundamental DPSO. Therefore, it is called DPSO-HSS. The detailed steps of the improved algorithm are summarized as follows.
**Algorithm 1**: DPSO with hybrid search strategies1. Generate the initial particle swarm that satisfying the conditions. Initialize *pBest_i_*, *gWorst*, and *gBest*. Initialize observation parameters Tpb*_i_* and *Txb_i_*. Initialize the solution set P_G_. 2. Calculate the fitness function value of each particle and update *pBest_i_*, *gWorst*, *gBest*, and Tpb*_i_*. Replenish the set P*_G_* with the good solutions that have been eliminated. 3. Update the velocity and position of the particle according to (16), (9), and (10). Determine whether the number of iterations *t* is larger than *t*_L_. If so, go to Step 4. Otherwise, go to Step 5. 4. Initiate the local search strategy. 5. Determine whether Tpb*_i_*> *u* and Txb*_i_*> *h* are both valid. If so, update the position of the particle according to (21). Otherwise, go to Step 6. 6. Constrain the particle position according to the constraint condition and update the parameter Txb*_i_*. 7. Do boundary treatment for particle velocity. 8. Output *gBest*. Determine whether the termination conditions are met. If so, end the optimization. Otherwise, *t* = *t* + 1, and return to Step 2.

## 4. Numerical Results and Analysis

In this section, several examples are presented to compare the performance, effectiveness and robustness of the DPSO-HSS algorithm and some contrast algorithms.

### 4.1. Simulation 1: Function Optimization Tests

The algorithms tested include the proposed DPSO-HSS algorithm, RDPSO algorithm in [13], and NPSOWM algorithm in [14].

The five typical functions used to compare the performance of each algorithm are shown in Table 1. The test is to minimize the function value within a specified range of dimensions and variables. In order to compare the effects of each algorithm under the same conditions, the test of each function is run several times and its statistical results are calculated for comparison.

The three statistical characteristics as evaluation criteria are:The mean value of multiple simulation results;The variance of multiple simulation results;The minimum value of multiple simulations.

The population size NP, the iteration times *T*, the dimension *D*, the learning factors *c*_1_ and *c*_2_, and the inertia weight *w* of the three algorithms are the same. The simulation results are shown in Table 2.

Among the five test functions, the proposed algorithm outperforms the other two algorithms in the mean and variance of Ackley, Rastrigin, and Sphere. In Rosenbrock’s test, DPSO-HSS has a better variance. In Griewank’s test, the test results of DPSO-HSS and RDPSO are close to the same. Considering the test results of the above five functions comprehensively, the proposed DPSO-HSS algorithm performs well in both mean and variance compared with the RDPSO and NPSOWM algorithm. The low mean indicates the excellent global search ability and convergence effect of the algorithm, whereas the low variance indicates that the stability of the algorithm results in multiple runs.

The time complexity of each test function is O (*D*), where *D* is the number of dimensions. By substituting this result into the optimization of the DPSO-HSS algorithm, the time complexity of DPSO-HSS can be calculated as follows
(22)$$T_{DPSO-HSS}=O\left[tp^{2}q+tpD\right]$$
where *D* represents the number of dimensions, *p* is the swarm population, *q* denotes the number of dimensions randomly selected from all *D* dimensions in (16), and *t* is the number of iterations. It can be seen that, compared with the general PSO algorithm, the improved algorithm has higher time complexity.

### 4.2. Simulation 2: Application in Planar Array Thinning with Constraints

The algorithms involved in the simulation include the proposed DPSO-HSS algorithm, RDPSO algorithm in [13], NPSOWM algorithm in [14], and MOPSO-CO algorithm in [4].

Consider a planar array consisting of 20 × 20 elements with equal spacing *d* = 0.5*λ* as an initial layout. *u* and *v* are both within the range [−1, 1] with a scanning step of 0.01. The main beam direction is (*θ*, *φ*) = (0°, 90°), that is, (*u*, *v*) = (0, 0). The filling rate is 50%, so there are 200 elements in the “ON” state. The population size NP = 100, iteration times *T* = 500, and dimensions *D* = 400 are the same for all algorithms used.

The convergence curve of PSLL using four algorithms for array thinning is shown in Figure 4. At the early stage of optimization, the DPSO-HSS algorithm does not have more impressive search efficiency than other algorithms, but it maintains good global searching ability with the guiding of comprehensive learning strategy and aims at maintaining the high diversity of the population and the correlation between the particles. At the later stage of optimization, the curve of the proposed algorithm shows the clear change process with local search strategy before reaching the optimal PSLL results. The trend of DPSO-HSS corresponds to the configured search strategy and meets the ideal expectation.

The elements distribution and beam pattern diagram of the optimized array of DPSO-HSS algorithm are shown in Figure 5 and Figure 6, respectively. In Figure 5, the black triangles represent the array elements in the “ON” state, and the number of these elements is 200, which meets the filling rate of 50%. The four angular array elements in the array are all in the “ON” state because the array aperture must be unchanged. On the other hand, the direction of the main beam is (*u*, *v*) = (0, 0) with the normalized amplitude of 0 dB, and the PSLL is −18.32 dB. These results mean the result of the array thinning design is correct.

We can see the PSLL performance in details in Figure 7, illustrating the novelty of the proposed DPSO-HSS algorithm, which plots the *u*-cut of the beam pattern diagram of a thinned array optimized by four kinds of algorithms. The main lobe width of the beam pattern of each algorithm is identical. The PSLL of the DPSO-HSS algorithm is −18.32 dB, whereas those of RDPSO, NPSOWM, and MOPSO-CO are −17.15 dB, −17.42 dB, and −17.66 dB, respectively. Compared with the other three algorithms, the PSLL of DPSO-HSS is decreased by 6.82%, 5.17%, and 3.74%, respectively.

### 4.3. Simulation 3: Beam Pattern of The Optimized Array with Different Main Beam Positions

In the above example, we set the main beam to a normal direction. Reference [21] proved that the main beam direction configuration has no impact to PSLL in unequally spaced linear antenna array thinning. In fact, this conclusion could also be obtained for planar array thinning. The *v*-cut of the radiation beam pattern diagram is shown in Figure 8 with different main beam directions, i.e., (*θ*, *φ*) = (0°, 90°), (*θ*, *φ*) = (30°, 45°), and (*θ*, *φ*) = (60°, −30°). The PSLLs are −18.32 dB, −18.30 dB, and −18.31 dB, respectively. The difference is less than 0.1 dB, which is consistent with the conclusion in [21].

### 4.4. Simulation 4: Synthesis Results of Four Algorithms for Large Planar Thinned Arrays

The array thinning effect may be affected by both array aperture and array filling rate. Table 3 records the PSLL and 3 dB bandwidth of the radiation beam pattern of thinned arrays optimized by each algorithm under different array apertures and filling rates. Each result is the mean value of five iterations running under the same simulation conditions.

The results of PSLL under different filling rates show that the filling rate does have a certain influence on the optimization effect of thinned array and the optimization effect will be reduced if the number of open elements is too many or too few. The PSLL is improved when the filling rate is a constant of 60% and the aperture is changed from 5λ to 10λ, which indicates that the aperture also has a certain influence on the performance of array thinning. As shown in Table 3, the 3 dB bandwidth of the main lobe is mainly affected by the aperture, and the larger the array aperture is, the smaller the 3 dB bandwidth is.

## 5. Conclusions

A novel optimization algorithm for large array thinning based on DPSO with hybrid search strategies is proposed to improve the performance of large planar array thinning. The proposed algorithm, named DPSO-HSS, utilizes a global learning strategy to improve the diversity of populations at the early stage of optimization. A dispersive solution set and the gravitational search algorithm are used during particle velocity updating. Then, a local mutation strategy is employed in the later stage of optimization so that the local convergence is enhanced by continuous search around the best position of the particle. Several of the above-mentioned representative examples of large planar arrays thinning are provided to demonstrate the effectiveness and robustness of the DPSO-HSS algorithm. The brief comparison results are shown in the Table 4.

## Figures and Tables

**Figure 1 sensors-22-07656-f001:**
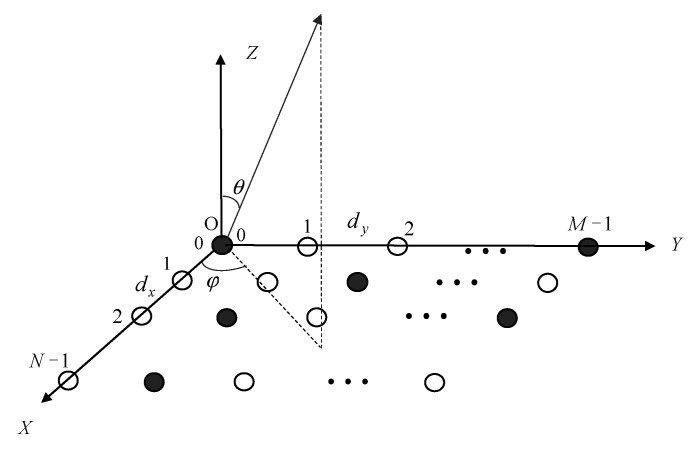
A planar array with some elements “ON” and the others “OFF”.

**Figure 2 sensors-22-07656-f002:**
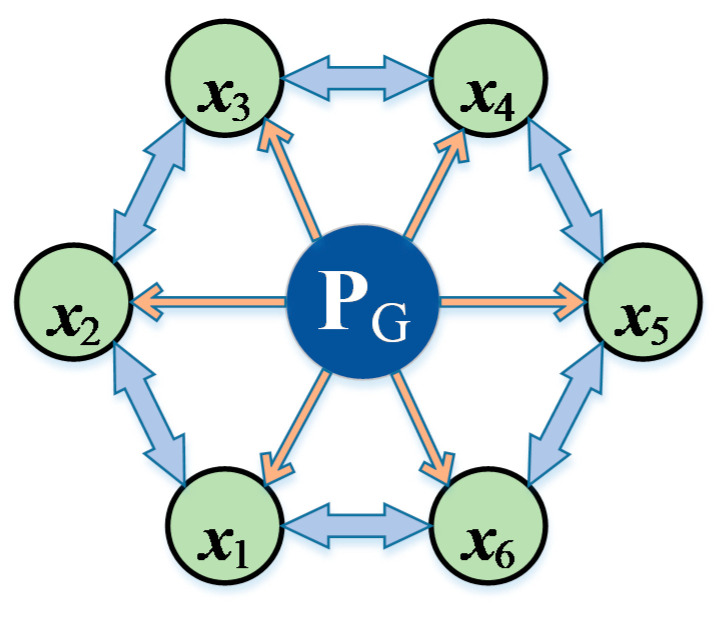
Schematic representation of the ring topology and the solution set P*_G_*.

**Figure 3 sensors-22-07656-f003:**
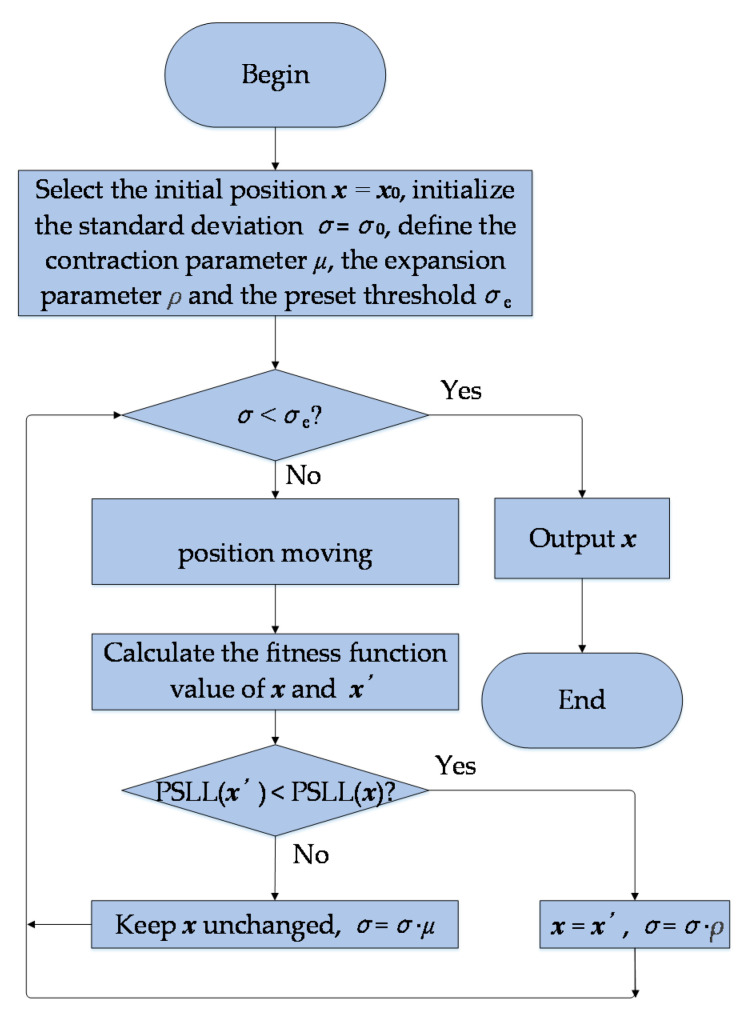
The flowchart of the local search strategy.

**Figure 4 sensors-22-07656-f004:**
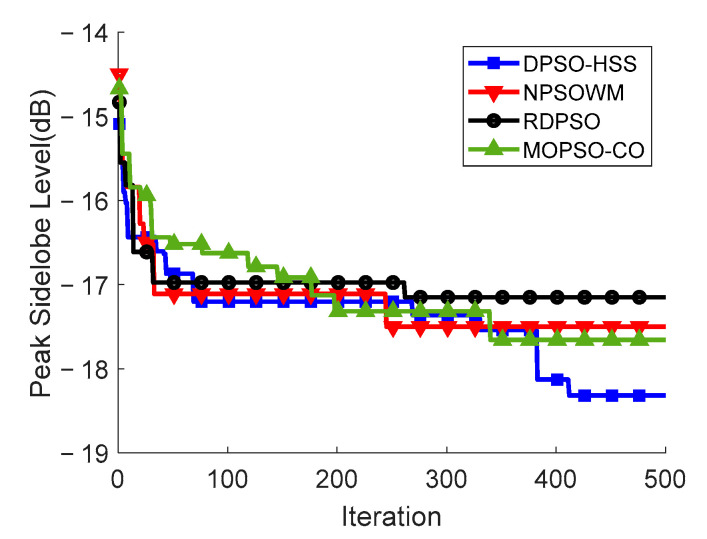
The convergence curve of PSLL using four kinds of algorithms in thinning the 10λ diameter array.

**Figure 5 sensors-22-07656-f005:**
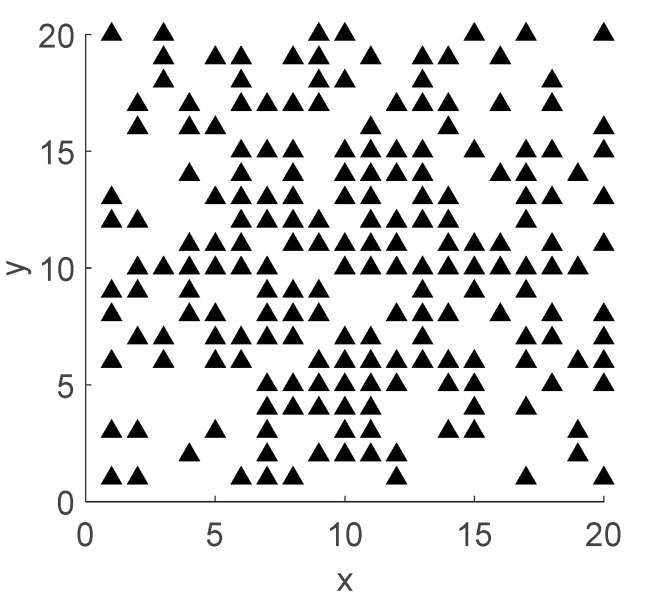
The array elements’ distribution of an optimized array using the DPSO-HSS algorithm.

**Figure 6 sensors-22-07656-f006:**
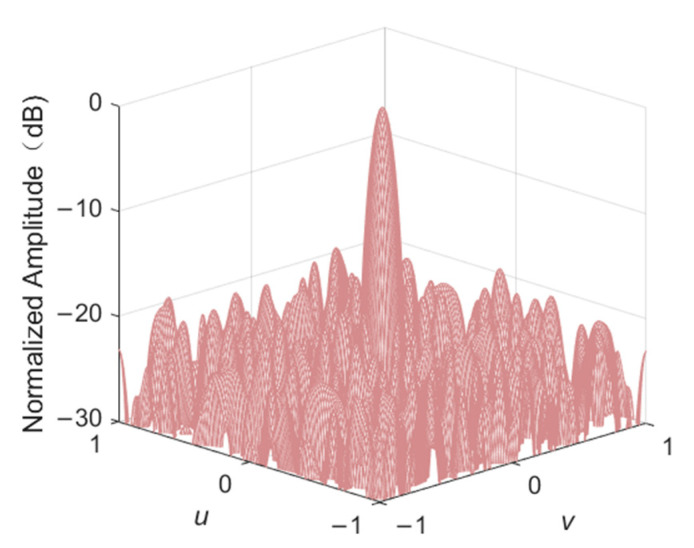
The radiation beam pattern diagram of an optimized array using the DPSO-HSS algorithm.

**Figure 7 sensors-22-07656-f007:**
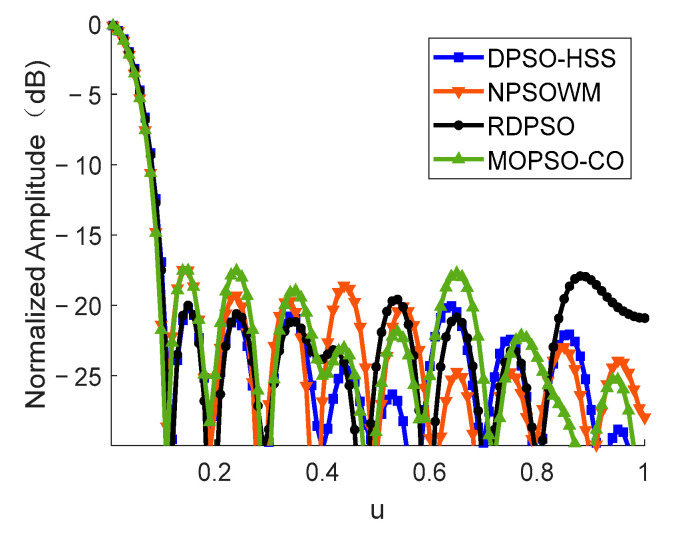
The *u*-cut of the beam pattern diagram of a thinned array optimized by each algorithm.

**Figure 8 sensors-22-07656-f008:**
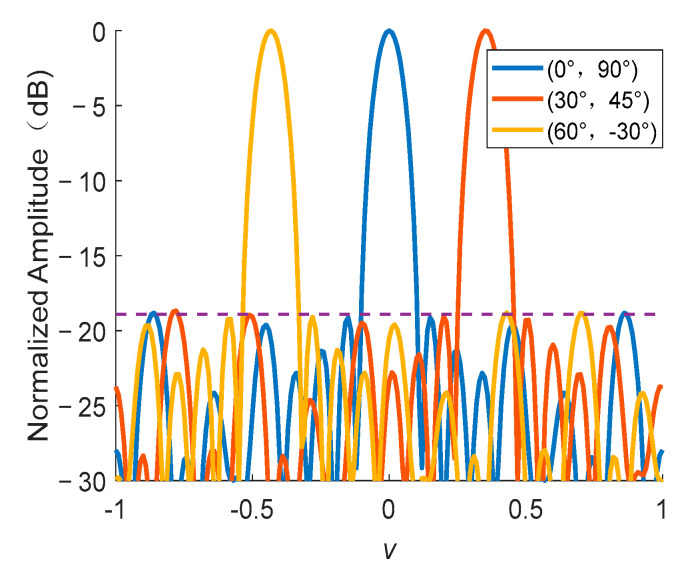
The *v*-cut of the radiation beam pattern diagram of the optimized array different main beam positions.

**Table 1 sensors-22-07656-t001:** Description of test functions and settings of simulation conditions.

Name	Range	Dimension	Minimum	Population	Runing time	Number of Iterations
Ackley	−30 ≤ *x_i_* ≤ 30	10	0	150	50	500
Rastrigin	−10 ≤ *x_i_* ≤ 10	10	0	200	50	1000
Sphere	−10 ≤ *x_i_* ≤ 10	30	0	100	100	500
Rosenbrock	−30 ≤ *x_i_* ≤ 30	10	0	200	50	1000
Griewank	−30 ≤ *x_i_* ≤ 30	10	0	100	100	500

**Table 2 sensors-22-07656-t002:** Statistical properties of function tests using different algorithms.

Function	Statistical Properties	DPSO-HSS	RDPSO [13]	NPSOWM [14]
Ackley	Mean	0.0096	2.8385	2.7257
	Variance	0.0005	1.2229	0.2512
	Minimum	0.0024	6.9407 × 10^−5^	1.4062
Rastrigin	Mean	7.5486	11.8201	32.1085
	Variance	9.4548	25.2794	98.8685
	Minimum	2.0026	3.9798	13.7634
Sphere	Mean	1.2493	2.1050	5.0407
	Variance	0.2234	1.8791	1.7301
	Minimum	0.5369	0.3208	2.2356
Rosenbrock	Mean	3.8523	2.3869	39.3544
	Variance	0.6871	7.4421	2.0763 × 10^3^
	Minimum	1.3338	1.3120 × 10^−13^	6.9166
Griewank	Mean	0.0801	0.0764	0.3513
	Variance	0.0027	0.0031	0.0104
	Minimum	0.0353	1.9080 × 10^−6^	0.1228

**Table 3 sensors-22-07656-t003:** Synthesis results obtained by four algorithms for large planar thinned arrays.

Aperture Diameter (×λ)	Fill Factor (%)	PSLL (dB)				3 dB Beamwidth (*u*)
		DPSO-HSS	RDPSO [13]	NPSOWM [14]	MOPSO-CO [4]	
5	100	−12.97	−12.97	−12.97	−12.97	0.179
5	90	−15.46	−15.25	−15.31	−15.38	0.182
5	80	−16.23	−15.83	−16.07	−16.12	0.185
5	70	−16.84	−16.48	−16.53	−16.70	0.181
5	60	−15.90	−15.59	−15.76	−15.64	0.180
6	60	−16.72	−16.49	−16.62	−16.55	0.159
7.5	60	−16.81	−16.45	−16.78	−16.57	0.125
9	60	−17.19	−16.77	−17.02	−16.99	0.105
10	60	−17.43	−16.98	−17.14	−17.23	0.094
10	50	−18.01	−17.45	−17.66	−17.89	0.096
10	40	−16.84	−16.41	−16.53	−16.59	0.095
10	30	−15.45	−15.06	−15.18	−15.13	0.095
10	20	−13.01	−12.83	−12.99	−12.92	0.092

**Table 4 sensors-22-07656-t004:** Comparison of the proposed algorithm and other algorithms in key performance metrics.

Metrics	DPSO-HSS	MOPSO-CO [4]	RDPSO [13]	NPSOWM [14]
Efficiency	Normal	Normal	Good	Good
Stability	Good	Normal	Good	Normal
PSLL	Good	Normal	Normal	Normal

## Data Availability

Not applicable.

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
