# Peer review of "Constrained Planar Array Thinning Based on Discrete Particle Swarm Optimization with Hybrid Search Strategies"

_sensors, 2022, doi:10.3390/s22197656_

Round 1

Reviewer 1 Report

In this work an optimization algorithm for large array thinning based on DPSO with hybrid search strategies is proposed to improve the performance of large planar array thinning. I believe that the present form of the manuscript needs some improvements prior to publication, thus I have to propose “Reconsider after minor revision”. The authors are encouraged to submit a revised version of their manuscript, taking into consideration the following minor issues:

 1. Authors are recommended to improve the state of the art analysis by discussing more related works. The related work discussed should be from more recent years like 2021, 2022.

 2. Authors are recommended to clearly demonstrate the motivation behind this work.

 3. Add a comparison table in the manuscript summarizing key performance metrics to compare the proposed work with recently published work (preferably from the years 2020-2022) in order to support the claim of authors that the proposed methodology is a novel method.

4. Authors are suggested to proof read the manuscript thoroughly for language and grammatical mistakes or use any professional tool for this purpose.

Reviewer 2 Report

1. In 3.2.1. global learning strategy (line 116) PG is mentioned. Is PG similar to the experience pools so it continuously generated as the algorithm iterates? Does it contain a priori initial parameters? If not, how to set the initial location and velocity parameters of the population before iteration? 

2. In the first test case, (simulation 1), can the author give a brief quantitative analysis of the time complexity and space complexity of the proposed algorithm DPSO-HSS?

3. Have multiple experiments been conducted in the second test case (simulation 2)? It is essential to eliminate the error caused by the randomness of the swarm intelligence algorithm itself through multiple independent repeated experiments.

4. Introduction section should be improved. The authors are suggested to add a brief introduction providing a background of swarm optimization algorithm applications to sensors, antennas, etc. In below appropriate suggestions are provided which can improve the quality of the introduction section. 

10.1109/TSMCC.2010.2054080

10.1109/TAP.2007.891552

10.1109/TIE.2021.3063873

10.1109/TAP.2005.856339

10.3390/s20164460

Reviewer 3 Report

In this paper, s a novel optimization algorithm for large array thinning. The algorithm is based on Discrete Particle Swarm Optimization(DPSO) integrated with some different search strategies. The idea is good. Following are my comments: 

1. Paper organization is missing. 

2. Better if authors can show its capacity with optimized results in real time problem. 

Round 2

Reviewer 2 Report

can be accepted